# Facile Fabrication of Eucommia Rubber Composites with High Shape Memory Performance

**DOI:** 10.3390/polym13203479

**Published:** 2021-10-11

**Authors:** Lin Xia, Jiafeng Meng, Yuan Ma, Ping Zhao

**Affiliations:** Key Laboratory of Rubber-Plastics, Ministry of Education/Shandong Provincial Key Laboratory of Rubber-Plastics, School of Polymer Science and Engineering, College of Materials Science and Engineering, Qingdao University of Science and Technology, Qingdao 266042, China; xialin@qust.edu.cn (L.X.); mengjiafengz@163.com (J.M.); ma1106yuan@163.com (Y.M.)

**Keywords:** Eucommia rubber, ZDMA, shape memory, composite

## Abstract

We processed a series of shape memory Eucommia rubber (ER) composites with both carbon–carbon and ionic cross-linking networks via a chemical cross-linking method. The influence of the carbon–carbon cross-linking and ion cross-linking degree of ER composites on curing, mechanical, thermal, and shape memory properties were studied by DSC, DMA, and other analytical techniques. Dicumyl peroxide (DCP) and zinc dimethacrylate (ZDMA) played a key role in preparing ER composites with a double cross-linking structure, where DCP initiated polymerization of ZDMA, and grafted ZDMA onto polymer molecular chains and cross-linked rubber molecular chains. Meanwhile, ZDMA combined with rubber macromolecules to build ionic cross-linking bonds in composites under the action of DCP and reinforced the ER composites. The result showed that the coexistence of these two cross-linking networks provide a sufficient restoring force for deformation of shape memory composites. The addition of ZDMA not only improved the mechanical properties of materials, but also significantly enhanced shape memory performance of composites. In particular, Eucommia rubber composites exhibited outstanding mechanical properties and shape memory performance when DCP content was 0.2 phr.

## 1. Introduction

Shape memory polymer (SMP) material refers to the polymer whose shape can be changed and fixed under some external conditions [1], such as heat, light, and electricity, etc. [2,3,4,5]. These polymer materials return to their original shape, and are used in many areas, e.g., biomedical treatment [6,7,8,9], aerospace [10], intelligent textiles [11,12], sensors and, self-repairing materials, etc. [13,14,15,16]. 

The thermal-induced SMP is a kind of typical shape memory polymer in which the cross-linked structure and/or the entanglement structure of molecular chains retain the permanent shape of the polymer [17,18,19,20,21]. When the temperature of SMP is lower than the transition temperature (T_trans_), the reversible domain stays in a hardened state. When it is heated to or higher than T_trans_, the reversible domain begins to soften, and movement and slip of molecular chains occur under the external force, leading to the change of shape. When cooled to or lower than T_trans_, the movement of molecular chains decreases and micro-Brownian motion weakens, which renders a certain deformation of molecular chains [22]. When heated again to the transition temperature of the reversible domain in the polymer, the molecular chains of the reversible domain are free again. In this way, the polymer reverts to its permanent shape, leading to achievement of the shape memory cycle [23,24].

Generally, there are carbon–carbon cross-linking bonds in the majority of shape memory polymers prepared by the cross-linking method, because the carbon–carbon cross-linking bond has a large bond energy and stability. However, many shape memory polymer composites prepared by carbon–carbon cross-linking have some shortcomings, such as insufficient mechanical strength and poor recovery rate [25]. An ionic bond is a type of electrovalent bond formed by the electrostatic attraction between oppositely charged ions in composites. The advantages of ionic cross-linking bonds include electrostatic attraction of ionic bonds, which can make the polymer cross-link at room temperature, and the ionic bonds are untied (dispersed or disconnected) when the temperature increases. Thus, the material performance can be optimized [26,27], and the ionic cross-linking bond was used in the preparation of polymer shape memory materials in this study.

We designed a novel composite with both carbon–carbon cross-linking bonds and ionic bonds using Eucommia rubber (ER) as the basic material. ER is mainly composed of trans-1,4-polyisoprene, which is the isomer of nature rubber cis-1,4-polyisoprene from Hevea brasiliensis. ER can be used as a type of shape memory material after being appropriately cross-linked [28,29,30,31]. The introduction of ionic cross-linking depends on the implementation of ZDMA, which is a typical ionic compound. It can be used as a rubber reinforcement, and also reacts with the rubber matrix [32,33]. Generally, ZDMA undergoes a high-temperature process in the rubber system and produces three structures in rubber composites: (a) it exists in the form of ZDMA, (b) it forms polymerized ZDMA (PZDMA) products, and (c) it generates a PZDMA-graft-rubber structure [31,34]. Some studies have been conducted on rubber and ZDMA, such as the research work by Qinghong Fang [35]. There are also some studies on the recoverable strain in shape memory behavior of rubber [36,37].

Here, we explored ZDMA’s excellent reinforcing effect and reactive activity in the ER matrix and constructed a double cross-linking system in this manuscript. The contribution of ionic cross-linking and covalent cross-linking was calculated separately in the whole cross-linking system. The change of composites’ strain under different loads was investigated intuitively, which can be used to directly observe the influence of the cross-linking density on the deformation of composites. The present study provides a new approach to prepare SMP composites and optimize their microstructure. Through testing, the composites had both traditional carbon cross-linking bonds and ion cross-linking bonds and had excellent mechanical properties and shape memory properties.

## 2. Materials and Methods

ER was extracted from Eucommia bark in our lab. ZDMA was purchased from Chengdu Organic Chemicals, Chinese Academy of Science (Chengdu, China). The other chemicals were used as received without any further treatment.

ER composites were prepared in a high-temperature mill at 90 °C. The ER composite formulations with various ratios are shown in Table 1, and the preparation diagram of composites is shown in Figure 1.

The cure characteristics of composites were tested according to ASTM D-2084-07. The fracture surface was observed by field-emission scanning electron microscope (SEM, JEOL JSM-6700F, Tokyo, Japan).

The cross-link density of samples was calculated according to Equations (1) and (2):− [ln (1 − φ_r_) + φ_r_ + χφ_r_^2^] = V_0_n (φ_r_^1/3^ − φ_r_/2)(1)
(2)$$\varphi_{r}=\frac{m_{2}\rho_{1}}{m_{2}\rho_{1}{+m}_{2}\rho_{2\;}{-\;m}_{1}\rho_{2}}$$
where φ_r_ is the volume fraction of swollen ER, and χ is the Flory–Huggins polymer–solvent interaction term [38]. m_1_ and *m*_2_ are the mass of swollen samples before and after drying, respectively. V_0_ is the molar volume of toluene, n is the number of active network chain segments per unit volume, and *ρ*_1_ and *ρ*_2_ are the density of ER and toluene, respectively.

These samples mentioned above were then placed into toluene/trifluoroacetic acid (TFA) for 120 h and into toluene at 60 °C for 72 h. After swelling samples to break the ionic cross-links, we repeated 3 times to swell the samples with toluene for 6 h to remove the residual TFA in the swelled samples. Next, the surface solvent was removed, and the sample was weighted again. Finally, the covalent and ionic cross-link density of samples was calculated again [39].

DSC measurement was carried out using a DSC-Q20 (TA Instruments, Newcastle, DE, USA). The temperature and enthalpy were calibrated with an indium standard. Samples with a mass of 6~10 mg were maintained at 100 °C for 3 min to eliminate their thermal histories before they were cooled to −50 °C at a speed of 10 °C∙min^−1^. Samples were subsequently heated to 100 °C at a speed of 10 °C∙min^−1^. The degree of crystallinity (X_c_) for samples was calculated using Equation (3):(3)$$X_{c}=\frac{∆H_{m}}{∆H_{m}^{*}}\;\times \;100\%$$
where ΔH_m_ and Δ$H_{m}^{*}$ are melting enthalpy of the sample and its melting enthalpy (ca. 186.8 J·g^−1^ for ER [40]). The dynamic mechanical analysis was carried out using a DMA Q800 (TA Instruments, USA) in the “controlled force” mode. The test samples were cut into rectangular shapes with a thickness of 2.0 mm, a width of 4.0 mm, and a length of 30.0 mm. The initial clamp gap was set to 5.0 to 10.0 mm. The sample was maintained isothermally at 100 °C for 5 min (ε_0_). Then, a 2 N stress was applied at a speed of 0.2 N∙min^−1^ to stretch the sample, which was followed by cooling to −50 °C to freeze the crystalline domain in the composites (ε_1,load_). Finally, the load was removed (ε_1_) at a speed of 0.2 N∙min^−1^, and the sample was reheated to 100 °C and maintained isothermally for 15 min (ε_0,rec_). The heating and cooling rates were both set to 5 °C∙min^−1^. The shape fixity ratio (R_f_) and shape recovery ratio (R_r_) of composites were calculated using Equations (4) and (5) [30]: (4)$$R_{f}(0\to 1)=\frac{\varepsilon_{1}-\varepsilon_{0}}{\varepsilon_{1,\mathrm{load}}{-\varepsilon }_{0}}\times 100\%$$
(5)$$R_{r}(1\to 0)=\frac{\varepsilon_{1}{-\varepsilon }_{0,\mathrm{rec}}}{\varepsilon_{1}{-\varepsilon }_{0}}\times 100\%$$
where ε_1,__load_ represents maximum strain under stress, ε_1_ and ε_0_ are strain after cooling and stress removal, and ε_0,rec_ is the recovered strain. 

## 3. Results and Discussion

### 3.1. Properties of ER Composites

#### 3.1.1. Cure Characteristics of ER Composites

The effect of DCP on cure characteristics of ER composites is shown in Figure 2, where M_H_ and M_L_ represent maximum and minimum torque during the curing process, respectively. M_H_ showed a gradual increase on the addition of ZDMA, which is shown in Figure 2a. Meanwhile, M_L_ was unchanged. Thus, M_H_ − M_L_ also showed an increasing trend, implying an increase of cross-linking rates in composites [41]. Moreover, the torque difference became larger and larger with the increase of DCP. The value of M_H_ − M_L_ with ZDMA was much larger than that without ZDMA, which indirectly illustrated that the addition of ZDMA can significantly increase the cross-linking density of the composites. In Figure 2b, T_10_ represents the scorch time, and T_90_ is closely linked with optimum cure time. The curing time T_90_ of formula E was longer than formula A to D, indicating that ZDMA can shorten the curing time of composites. The curing time increased with peroxide DCP content when the amount of ZDMA was fixed. When the amount of DCP was 0.2 phr, the curing time was shortest at 160 °C. However, the scorch times of composites showed the opposite trend, which was shortened with the increase of DCP content. The short scorch time is a challenge for safe processing of rubber composites.

#### 3.1.2. Mechanical Properties of ER Composites

The addition of ZDMA has a great influence on the properties of rubber materials. In particular, the microstructure of composites changes significantly due to the addition of DCP. Generally, DCP mainly plays two important roles in the preparation of ER composites. First, it induces vulcanization of ER. Second, DCP enables ZDMA to generate free radicals, forming polymerized ZDMA (PZDMA) or terminating with free radicals of ER, resulting in the formation of rubber-graft-PZDMA [42]. From Figure 3, we note that the addition of ZDMA can significantly improve the mechanical properties of ER composites. Comparing sample A with ZDMA and sample E without ZDMA, the tensile strength and tear strength of composites increased to 31.2 MPa and 94.6 KN/m, respectively. However, the tensile strength, tear strength, 100% modulus, and elongation at break showed a decreasing trend with the increase of DCP dosage and fixed content of ZDMA. Thus, mechanical properties of composites decreased with the increase of DCP content, when the ZDMA content was kept constant. 

Then, we tested the tensile strain at different stresses and at higher deformation temperature (100 °C), in order to observe the influence of DCP content on the properties of composites (Figure 4). The tensile strain of composites decreased gradually with the increase of DCP content at a stress of 1 N (Figure 4a), 2 N (Figure 4b), 3 N (Figure 4c), and 5 N (Figure 4d), which is closely related to the increase of cross-linking density in composites. In particular, the tensile strain of sample A reached 222.7% at an external stress of 5 N. We also found that the larger the external force, the larger the tensile strain. Therefore, we can select the appropriate cross-linking degree to optimize the tensile strain according to the requirements of composites.

These mechanical properties above can infer that the mechanical properties are closely related to the microstructure of composites, especially the supramolecular network of composites, which contained cross-linking networks of ER, PZDMA, and graft-polymerization of ZDMA onto ER molecules [43].

#### 3.1.3. Swelling Cross-Link Density of ER Composites

Then, we investigated the types of cross-linking included in the microstructure of composites through swelling experiments. We immersed the composites in acid and toluene solvents, calculated the cross-linking degree, and analyzed the internal structure of the materials by observing the change of solvent color [44]. Given that ionic cross-links are obtained because of electrostatic interaction between the ion pairs, they can be destroyed by small polar molecules such as acid, chlorine, ethanol, and hydrogen ions [28,29,30,31,32]. Therefore, the mixture of toluene and trifluoroacetic acid (TFA) can destroy all the ionic cross-links in composites. In the test of cross-linking degree, the samples were first placed in toluene to calculate the total cross-links. Then, the swollen samples were directly transferred into the mixture of toluene/TFA to destroy ionic cross-links after calculation of total cross-linking degree in the toluene. The residual network was mainly constructed by covalent cross-links, which was determined by subtraction of ionic cross-link density from total cross-links. Sample A was dispersed in a mixture of toluene and TFA, and there was no bulk solid in the mixed solution after the swelling experiment (Figure 5a). This phenomenon indicated that there were no covalent cross-links in sample A, and the data of sample A also proved this point with no covalent cross-linking in composites when DCP content was 0.2 phr (Figure 5b). The color of samples B, C, and D was basically similar in Figure 5a, and no obvious change could be observed. Therefore, the above phenomenon indicated that the cross-links in sample A were destroyed by acid, and all of which were ionic cross-links. The samples B, C, and D were opaque and dark brown in the mixture of toluene and TFA. The cross-linking density of composites with different DCP content is shown in Figure 5, which gives an intuitive observation of the formation of a supramolecular network in ER composites. After calculations, we found that ionic, covalent, and total cross-linking degree indicated an increasing trend with the increase of DCP content. 

Thus, a schematic diagram is proposed to explain the internal structure of composites, indicating the effect of DCP and ZDMA in ER composites. As shown in Figure 6, the red star represents the aggregation of the ZDMA or PZDMA part, which can be combined with the polymer chain of ER. The disorderly dispersed lines in composites represent amorphous chain segments, and regularly arranged lines represent crystalline regions of ER composites. Green points represent cross-linking points in composites. From Figure 6, we note that ZDMA plays the role of cross-linking agent and reinforcing agent in composites, such that the addition of ZDMA has a great impact on the mechanical properties and shape memory properties of the material.

#### 3.1.4. DSC and DMA Analysis of Composites

The crystallization and cross-linking structure of materials play an important role in determining the mechanical and shape memory properties of composites. We performed DSC to test the thermal properties of composites. From the curves in Figure 7a,b, we found that the addition of ZDMA and the increase of DCP content made the crystallization temperature and melting point of the material move to the low temperature region. We also calculated the degree of crystallinity and melting temperature based on the values of enthalpy in DSC curves (Figure 7c,d). The degree of crystallinity and melting temperature decreased with the increase of DCP content, implying that the increase of cross-linking degree and the addition of ZDMA destroyed the crystallization of ER materials. The crystallinity of the composite was 21.6% without ZDMA, and when the amount of ZDMA was fixed at 5 phr, the crystallinity of the composite decreased from 20.7% to 17.8% with the increase of DCP, which greatly affected the mechanical properties of materials. 

Meanwhile, we examined the effect of ionic bond and cross-linking degree on the shape memory properties of ER composites. From Figure 8 and Table 2, we note that R_f_ and R_r_ of pure ER material were 95.30% and 90.12%, respectively. R_f_ and R_r_ of ER composites changed to 93.51% and 93.93% with the addition of ZDMA (5 phr), respectively. The change of R_f_ and R_r_ may be because of the change of the internal structure of composites. On the one hand, ionic bonds formed inside the polymer chains, which destroyed the crystallization of ER and affected R_f_ of composites. On the other hand, the type of cross-linking network structure was changed, including both traditional covalent cross-linking and ionic cross-linking, which is closely related to R_r_ of composites. When the amount of DCP was 0.2 phr, and ZDMA was not added, the composites had completely covalent carbon cross-linking bonds. However, when ZDMA was 5 phr, the internal section of the material was characterized with completely ionic cross-linking bonds (Figure 8b). R_f_ decreased gradually with the increase of the degree of cross-linking. This result validates the fact that the increase of cross-linking networks and the addition of ZDMA led to the destruction of crystallization, which resulted in the decrease of the shape fixing rate. In contrast, R_r_ slightly increased with the increase of DCP content. This phenomenon is attributed to the increase of the total cross-linking density in the system. The increase of cross-linking density endowed the composites with high elasticity, namely, entropy elasticity, which can make the material recover faster and in a more efficient manner. 

Figure 9 shows the shape memory process of composites. When the sample was maintained isothermally at 100 °C, the crystalline regions were melted, which provided the condition for the deformation of composites. A temporary shape was obtained under mechanical loading at temperatures above the T_trans_ of ER composites. Following the cooling process, a temporary shape was fixed with the removal of external force. Specimens returned to their original shapes when they were reheated again. Therefore, the melting crystallization zone provided the conditions for the deformability of ER composites, and the cross-linking network can provide the shrinkage force for the deformation of the composites during the entire deformation process. The schematic diagram reflects softening, deformability, fixing, and restoring of the crystalline parts in composites during the heating and electrification process. The lower half of Figure 9 is the image of the sample with the initial shape and temporary shape. We first heated and softened the sample with an initial length of ~4 cm, which could be stretched to ~16 cm. After soaking in hot water, the sample with a temporary length of 16 cm could recover to the original length of ~4 cm.

## 4. Conclusions

We employed ZDMA-induced ionic cross-linking to prepare a series of novel shape memory Eucommia rubber composites, which contained not only common carbon–carbon cross-linking networks, but also ionic cross-linking networks. We observed that DCP and ZDMA played key roles in preparing ER composites with a double cross-linking structure. The main function of DCP was to initiate ZDMA polymerization, and to graft ZDMA onto polymer molecular chains and cross-linked rubber molecular chains. Meanwhile, ZDMA had the following functions: on the one hand, it combined with rubber macromolecules to build ionic cross-linking bonds in composites under the action of DCP; on the other hand, it reinforced the ER composites. When the content of DCP was 0.2 phr, the composite was mainly composed of ionic cross-links. The addition of ZDMA not only improved the mechanical properties of materials, but also contributed significantly to the shape memory performance when the DCP content was 0.2 phr.

## Figures and Tables

**Figure 1 polymers-13-03479-f001:**
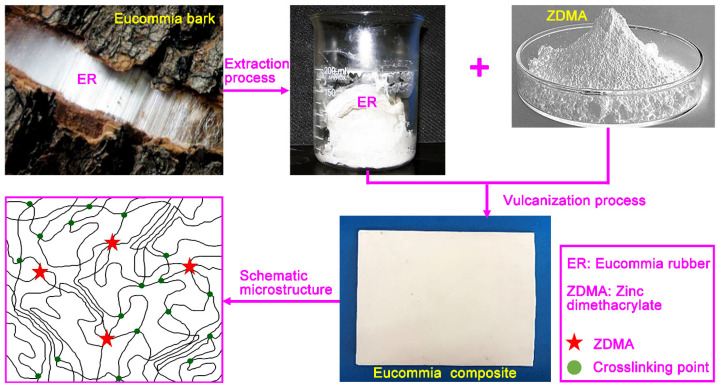
Schematic illustration of preparation for ER composite.

**Figure 2 polymers-13-03479-f002:**
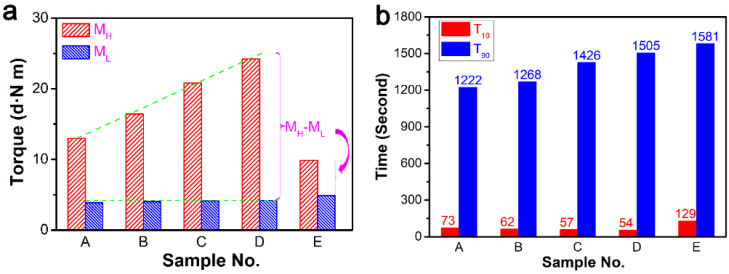
Effect of DCP on the cure characteristics of ER composites: (**a**) maximum torque (M_H_) and minimum torque (M_L_) during the molding process; and (**b**) the scorch time with 10 s (T_10_) and 90 s (T_90_).

**Figure 3 polymers-13-03479-f003:**
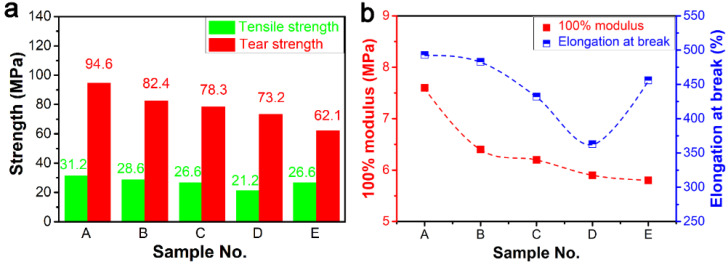
Effect of DCP dosage on the mechanical properties of the ER composites: (**a**) tensile strength and tear strength; (**b**) 100% modulus and elongation.

**Figure 4 polymers-13-03479-f004:**
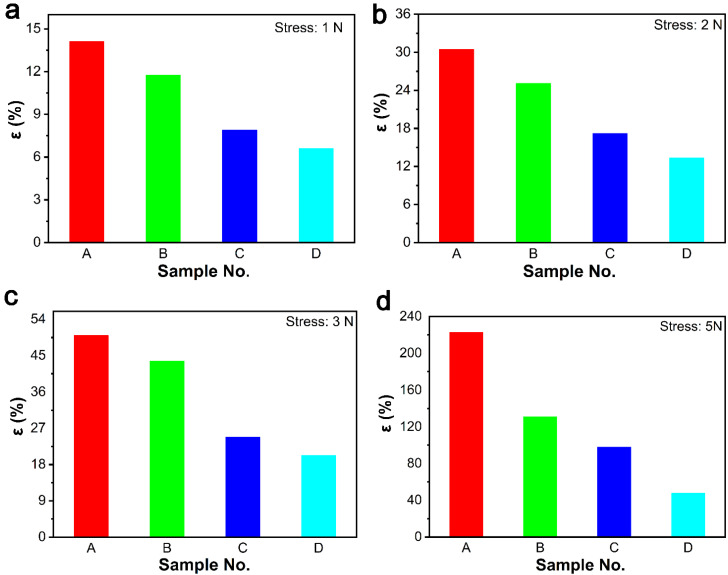
The strain of ER composites under different stresses at 100 °C: (**a**) 1 N; (**b**) 2 N; (**c**) 3 N; and (**d**) 5 N.

**Figure 5 polymers-13-03479-f005:**
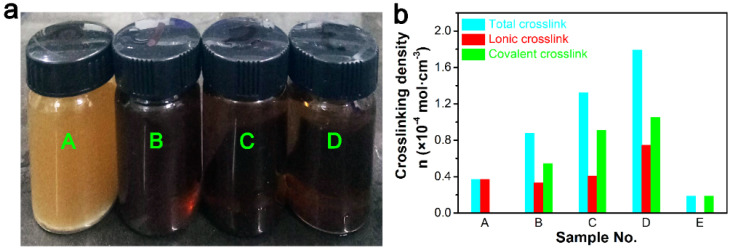
(**a**) Digital photographs of composites with different DCP content after dissolution/swell experiment, and (**b**) cross-link density of ER composites with different DCP content.

**Figure 6 polymers-13-03479-f006:**
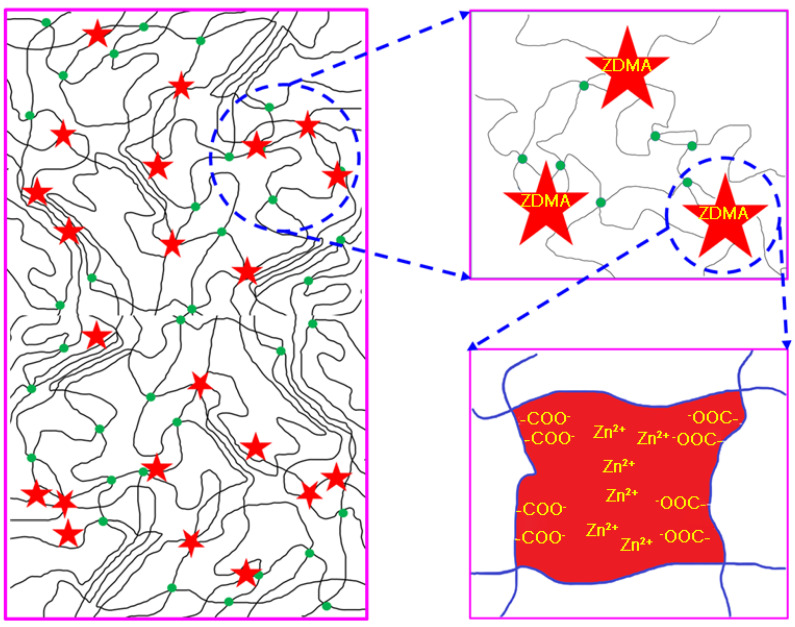
Schematic illustration of the microstructure in ER composites.

**Figure 7 polymers-13-03479-f007:**
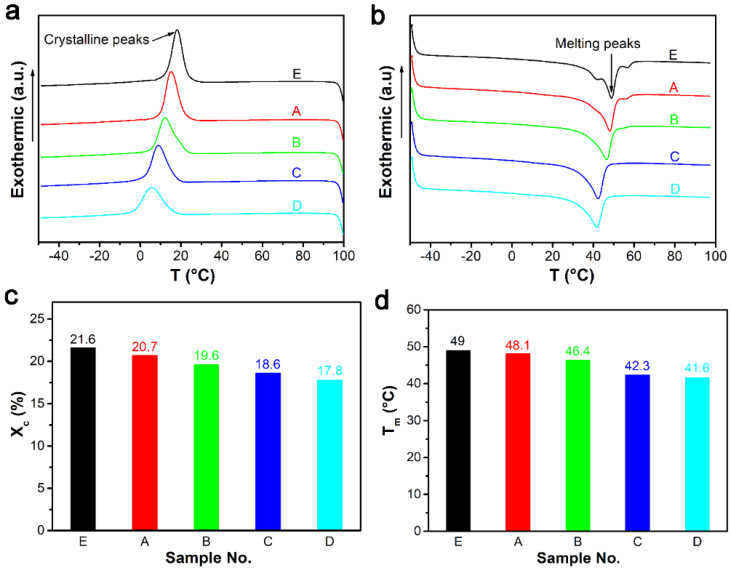
DSC curves of ER composites with different DCP content: (**a**) crystalline peaks of DSC curves; (**b**) melting peaks of DSC curves; (**c**) crystalline degree obtained from DSC curves; and (**d**) Tm obtained from DSC curves.

**Figure 8 polymers-13-03479-f008:**
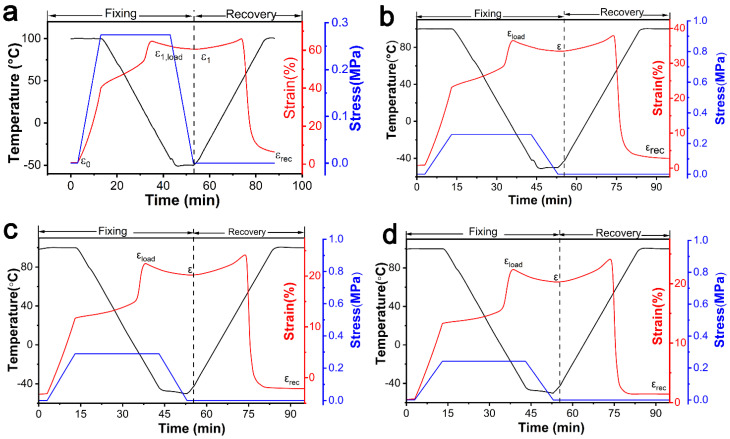
Shape memory performance of ER composites with different DCP content: (**a**) 0.2 phr; (**b**) 0.4 phr; (**c**) 0.8 phr; and (**d**) 1.2 phr.

**Figure 9 polymers-13-03479-f009:**
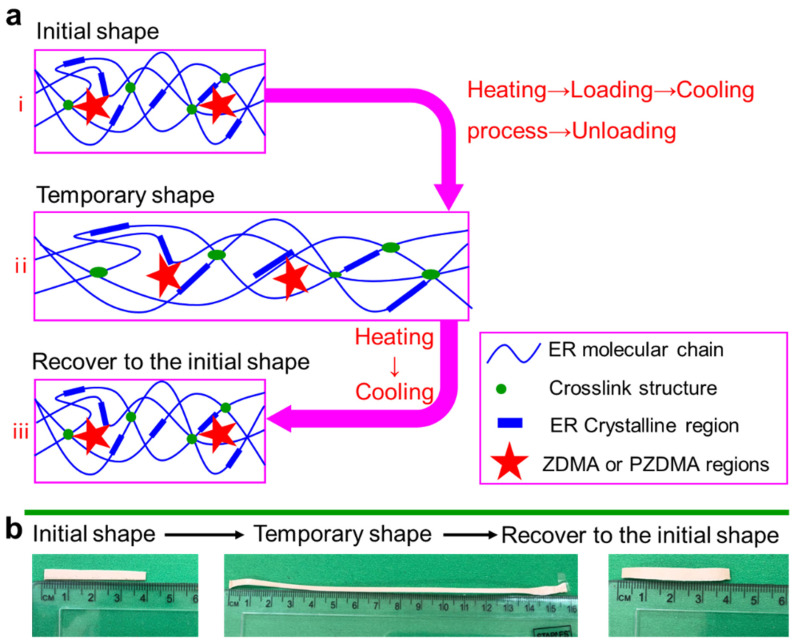
(**a**) Schematic diagram of the shape memory process of ER composites, and (**b**) physical diagram of shape memory process.

**Table 1 polymers-13-03479-t001:** Formulation of the ER composites.

Chemical (phr) ^a^	A	B	C	D	E
ER	100	100	100	100	100
ZDMA	5	5	5	5	0
Antioxidant 264	2	2	2	2	2
DCP	0.2	0.4	0.8	1.2	0.2

^a^ Parts by weight per hundred parts of rubber.

**Table 2 polymers-13-03479-t002:** Effect of the DCP dosage on the R_f_ and R_r_ values for SME in the composites.

Properties	A	B	C	D	E
Rf (%)	93.51 ± 0.30	91.65 ± 0.25	89.98 ± 0.28	90.66 ± 0.27	95.30 ± 0.26
Rr (%)	93.93 ± 0.28	94.33 ± 0.11	94.65 ± 0.20	95.58 ± 0.18	90.12 ± 0.30

## Data Availability

The data presented in this study are available on request from the corresponding author.

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
