# Peer review of "Facile Fabrication of Eucommia Rubber Composites with High Shape Memory Performance"

_polymers, 2021, doi:10.3390/polym13203479_

Round 1

Reviewer 1 Report

The phenomenon of shape memory is nothing new for composites. The authors attempted to prepare SMP composites and optimize its microstructure. They found both traditional carbon cross-linking bond and ion cross-linking bond and had excellent mechanical properties and shape memory properties. I suggest the authors include more recent studies in recent years in the reference section.

Author Response

Dear Editor and Reviewers,

Thank you for sending the reviewers’ comments concerning our manuscript entitled “Facile fabrication of Eucommia rubber composites with high shape memory performance” (Manuscript ID: Polymers-1379805). Those valuable comments are very helpful for improving our manuscript. We have considered the comments thoroughly and have made revisions which we hope could meet the requirements of the journal. Revised portion are highlighted in blue in the manuscript. The main corrections in the paper and the responses to the editor and reviewer’s comments are as following:

Reviewer: 1

Comments to the Author
The phenomenon of shape memory is nothing new for composites. The authors attempted to prepare SMP composites and optimize its microstructure. They found both traditional carbon cross-linking bond and ion cross-linking bond and had excellent mechanical properties and shape memory properties. I suggest the authors include more recent studies in recent years in the reference section.

Response: Thanks for your kindly comments. We have added recent studies in the reference section. We will strive for some innovative work in the future. Thank you for your advice.

Reviewer 2 Report

The manuscript entitled “Facile fabrication of Eucommia rubber composites with high shape memory performance” is approached to combine the two class of crosslinking in Eucommia rubber, namely, carbon-carbon and ionic bonds. The research was correctly proposed, and the introduction is concise and contents sufficient references to justify the intention of the work. The results are properly presented, and it is evident that this work was systematically carried out. The whole manuscript has a average quality, which can be enhanced. In this regard, there are some concerns of this reviewer:

  1. In the experimental was not neatly work. The first three paragraphs correspond to the instruction of the MDPI manuscript templated. Please correct it.
  2. The discussion is missing. Only, the results section (3) appears, but apparently the discussion is included there. So, the tittle should be changed to “results and discussion”.
  3. The discussion of the results indicates the knowledge in this topic of the authors, but this discussion is intuitive mainly, so references to support the inferences and conclusion of this study should be included.

Author Response

Dear Editor and Reviewers,

Thank you for sending the reviewers’ comments concerning our manuscript entitled “Facile fabrication of Eucommia rubber composites with high shape memory performance” (Manuscript ID: Polymers-1379805). Those valuable comments are very helpful for improving our manuscript. We have considered the comments thoroughly and have made revisions which we hope could meet the requirements of the journal. Revised portion are highlighted in blue in the manuscript. The main corrections in the paper and the responses to the editor and reviewer’s comments are as following:

Reviewer: 2
Comments to the Author
The manuscript entitled “Facile fabrication of Eucommia rubber composites with high shape memory performance” is approached to combine the two class of crosslinking in Eucommia rubber, namely, carbon-carbon and ionic bonds. The research was correctly proposed, and the introduction is concise and contents sufficient references to justify the intention of the work. The results are properly presented, and it is evident that this work was systematically carried out. The whole manuscript has a average quality, which can be enhanced. In this regard, there are some concerns of this reviewer:

  1. In the experimental was not neatly work. The first three paragraphs correspond to the instruction of the MDPI manuscript templated. Please correct it.

Response: Thanks for your kindly comments. We have revised the format of this manuscript. Please review it again.

  1. The discussion is missing. Only, the results section (3) appears, but apparently the discussion is included there. So, the tittle should be changed to “results and discussion”.

Response: Thanks for your kindly comments. We have revised the format of this manuscript and added “Results and discussion” part. Please review it again.

  1. The discussion of the results indicates the knowledge in this topic of the authors, but this discussion is intuitive mainly, so references to support the inferences and conclusion of this study should be included.

Response: Thanks for your kindly comments. We have added the relevant references in the “Results and discussion” part to support our conclusion. Thank you for your good advice.

Round 2

Reviewer 2 Report

Dear authors,

I don't have any comment about the revised version of the manuscript. so I recomend to accept your manuscript. 

Best wishes